# Clinical and Radiological Outcomes of Computer-Assisted versus Conventional Total Knee Arthroplasty at 5-Year Follow-Up: Is There Any Benefit?

**DOI:** 10.3390/jpm13091365

**Published:** 2023-09-08

**Authors:** Sergio Chávez-Valladares, Jose Maria Trigueros-Larrea, Sergio Pais-Ortega, Maria Antonia González-Bedia, Alberto Caballero-García, Alfredo Córdova, David Noriega-González

**Affiliations:** 1Orthopaedics and Traumatology Surgery, Hospital Clínico Universitario de Valladolid (HCUV), 47003 Valladolid, Spain; jtriguerosl@saludcastillayleon.es (J.M.T.-L.); spais@saludcastillayleon.es (S.P.-O.); magonzalezbed@saludcastillayleon.es (M.A.G.-B.); 2Department of Anatomy and Radiology, Health Sciences Faculty, GIR: “Physical Exercise and Aging”, Campus Universitario “Los Pajaritos”, University of Valladolid, 42004 Soria, Spain; alberto.caballero@uva.es; 3Department of Biochemistry, Molecular Biology and Physiology, Health Sciences Faculty, GIR: “Physical Exercise and Aging”, Campus Universitario “Los Pajaritos”, University of Valladolid, 42004 Soria, Spain

**Keywords:** navigation, robotics, TKA, computer-aid

## Abstract

Computer-assisted total knee arthroplasty (CAS) remains controversial. Some authors defend that its improvement in knee alignment and positioning positively impacts arthroplasty survival rates, while others have stated that there is minimal or no difference compared to the conventional technique (cTKA). This paper features a retrospective, single-center, single-surgeon study, evaluating CAS surgery vs. regular cTKA in patients who consecutively underwent surgery between 2015 and 2017 (60 CAS patients vs. 59 cTKA). Data collection includes surgery duration, length of stay, blood loss and both preoperative and postoperative clinical outcome evaluation using WOMAC, SF-12, Forgotten Joint Score and VAS. Radiograph evaluation includes the tibiofemoral angle, posterior condylar offset and its ratio, and notching frequency and measurement. A total of 119 patients were included: 60 in the CAS group and 59 in the cTKA. Mean follow-up was 5.61 years (Max 7.83–Min 5.02 years). No clinically relevant preoperative differences were observed between the groups. Postoperatively, both groups showed similar functional results (WOMAC, SF-12, FJS, KSS, and VAS) with similar complication rates. The CAS group had an increased surgery time by a mean of 12 min (107.02 ± 15.22 vs. 95.32 + 13.87; *p* = 0.00) as well as a higher notching frequency and size (40% vs. 13.60%; *p* = 0.013; 1.239 mm ± 1.7604 vs. 0.501 mm ± 1.4179; *p* = 0.031). CAS obtained similar functional, radiological, and complication rates to cTKA at the expense of increasing surgery time and notching frequency and size.

## 1. Introduction

Total knee arthroplasty (TKA) is an excellent surgical treatment for patients who have end-stage osteoarthritis [1] with an implant survivorship of over 97% at 10 years [2]. Computer-assisted TKA (CAS) was developed to aid in the surgical procedure. Its defendants assert that it improves implant alignment and positioning, thereby improving patient function and implant survivorship. Early studies showed promising results, framing CAS as the next revolution in TKA [3,4,5,6,7], particularly for outliers [8]. Meanwhile, posterior studies questioned its impact on clinical results [9,10,11] compared with conventional TKA (cTKA). This lack of solid results may have prevented wider implementation of this technology.

We designed a study comparing both TKA methods and asked whether CAS represents an improvement in the clinical results, complication rate, radiographic results, and survival rate of TKA components. We also compare the results in patients with postoperative coronal alignment > ±3° between both groups and their clinical behavior.

## 2. Materials and Methods

Between January 2015 and December 2017, a total of 119 TKA were electively performed by a single knee expert surgeon (T-JM) for primary or secondary arthritis. T-JM had already performed 50 CAS surgeries before the beginning of the study period. All cases were interviewed, and radiographs were taken between January and March 2023.

Patient inclusion criteria were age 20–85, BMI < 35, consent for primary knee arthroplasty and diagnosis of osteoarthritis. The exclusion criteria were revision surgery, TKA for acute/subacute post-traumatic osteonecrosis or rheumatoid arthritis.

All surgeries were performed using the parapatellar approach with a standard single paramedian incision. Both groups used a cemented posterior stabilized implant aiming for adjusted mechanical alignment 0 ± 3°. The navigation system used was the Orthopilot navigator system TKA 5.1 version (Braun-Aesculap, Tuttlingen, Germany). Regarding implants, for the conventional group, we use Stryker-Triathlon^®^ (Stryker, Mahwah, NJ, USA) posterostabilized cemented prostheses, while for the CAS group, the chosen implant was Braun-Aesculap VEGA^®^ (Braun-Aesculap, Tuttlingen, Germany) posterostabilized cemented.

Our CAS system is based on landmark acquisition using bone markings. The mechanical axes of the limb, femur and tibia are then determined by registering the center of rotation for the hip, trochlear notch and tibial spines as well as palpation of the medial and lateral malleoli to determine the center of the ankle. After all landmarks are registered, the tibial cut is performed first with a jig positioned using computer indications at 0° of varus and 0° of tibial slope. Afterward, the use of a distractor allows us to evaluate gaps in flexion and extension. Before any cut is made, the system makes it possible to virtually plan and achieve a correct balance. If correct alignment and balance are not feasible by bone cuts only, ligament release is performed in standard fashion. Once correct balance is achieved, the femoral cut is performed and both cuts are checked by a computer. Subsequently, all femoral, tibial and liner trials are inserted, and the range of motion (ROM), frontal alignment, and varus/valgus laxity are checked by applying manual stress. If needed, the liner is upscaled and further releases are performed.

The same postoperative protocol was used for all patients. Multimodal analgesia including opioids and cryotherapy was always applied. On day one after surgery, patients received a physical therapy session with active flexo-extension exercises, patella passive mobilization, isometric quadriceps/hamstrings exercises and initiate ambulation with crutches. The discharge criteria were as follows: VAS < 4, extension deficit less than 10°, flexion greater than 90° and ability to walk with crutches for more than 20 m. Once at home, patients followed an ambulatory rehabilitation regime with up to 2 sessions per week for at least 6 weeks plus a domiciliary rehabilitation program.

Radiological evaluation included preoperative Kellgren–Lawrence classification and hip–knee–ankle (HKA) angle measurement. The posterior condylar offset (PCO) was evaluated on lateral radiographs per Chang et al. [12] by measuring the maximum distance between the tangent of the femoral diaphysis posterior cortex and posterior condylar margin. The PCO ratio (PCOR) was calculated by dividing the PCO by the maximum distance between the posterior condylar border and the tangent of the femoral diaphysis anterior cortex. The depth of anterior femoral notching was measured as the distance between the anterior cortex line and the anterior cut line of the distal femur [13].

The surgery duration, length of hospital stay, blood loss and visual analog scale while walking (VAS) scores were compared between the groups. Functional and quality of life outcomes were measured to compare the preoperative and postoperative ROM, Forgotten Joint Score (FJS) [14], Knee Society Score (KSS) [15], Short Form Survey (SF-12, Mental (MCS) and physical (PCS) scores) [16], and Western Ontario and McMaster Universities Osteoarthritis Index (WOMAC) [17]. All were assessed by two authors not involved in the surgical interventions (CV-S, PO-S).

Navigation-related complications were identified. Survivorship was compared between the two groups by defining loosening or failure as the visual radiographic migration of one or more component(s), substantial wear or osteolysis at the last follow-up.

Data analysis was performed using IBM SPSS Statistics 26. The Levene test was used to assess the homogeneity of variance (constant variance). Postoperative results were compared using a two-sample Student’s *t*-test, with the assumption of homogeneity of variance used as appropriate, for quantitative and normal distributed variables, and a Mann–Whitney U-test and Chi-square test for no parametric variables. For continuous variables and the differences between the two means, 95% confidence intervals were calculated. Two-tailed values of *p* < 0.05 were considered significant.

## 3. Results

A total of 60 cases were performed with CA, and 59 underwent the conventional technique. The study group consisted of 80 women and 39 males with a mean age of 73 (range 54–85 years) at the time of index surgery with a mean follow-up of 5.61 years (Max 7.83–Min 5.02 years). Women’s preponderance may be explained by the demography of the local population. Demographic data are available in Table 1.

### 3.1. Preoperative Status

There were no differences in the preoperative evaluation of the KSS, SF-12, WOMAC functional or pain sections. However, the mean values for WOMAC stiffness (3.83 ± 1.659 vs. 4.8 ± 1.156; *p* = 0.024) and VAS score (Table 2) were not homogeneous in both groups, with higher values in the cTKA group. In addition, cTKA showed higher preoperative values for hemoglobin (14.357 ± 1.404 vs. 14.211 ± 1.5246; *p* = 0.01), hematocrit (42.825 ± 3.82 vs. 42.614 ± 4.3594; *p* = 0.02) and flexion ROM (113.08 ± 9.61 vs. 106.61 ± 12.368; *p* = 0.01). (Table 2).

### 3.2. Radiological Results

Preoperatively, there were similar values for both the cTKA and CAS groups with no significant differences between groups and with the following Kellgren–Lawrence (KL) grade frequencies: 0% Grade 1, 8.33% Grade 2, 58.33% Grade 3 and 33.33% Grade 4 for the CAS group; and 0% Grade 1, 13.56% Grade 2, 61.02% Grade 3 and 25.42% Grade 4 for the cTKA group (*p* > 0.05). The HKA angle showed slight differences in the coronal plane: 7.37 ± 6.06 vs. 6.45 ± 6.61 (*p* = 0.035). No other significant differences were observed in the preoperative evaluation.

Postoperatively, no differences were observed in alignment results. Nevertheless, there was an increase in notching frequency and size (40% vs. 13.60%; *p* = 0.013; 1.239 mm ± 1.7604 vs. 0.501 mm ± 1.4179; *p* = 0.031) and PCOr (0.4307 ± 0.0594 vs. 0.4259 ± 0.0428; *p* = 0.003) in the CAS group. (Table 3).

### 3.3. Postoperative Results

There was an increase in surgery time by a mean of 12 min in the CAS group (107.02 ± 15.22 vs. 95.32 ± 13.87; *p* = 0.00). A comparison of the two groups after surgery showed interesting results in terms of ROM. The cTKA group showed a greater flexion ROM (118.17 ± 14.08 vs. 112.63 ± 11.647; *p* = 0.023), whereas the CAS group demonstrated better extension (0 ± 0 vs. −0.63 ± 1.884; *p* = 0.011). Furthermore, in terms of functional results, the CAS group showed better performance on the WOMAC pain (2.27 ± 2.09 vs. 3.78 ± 2.748; *p* = 0.01) and stiffness subscale (1.17 ± 1.416 vs. 2.02 ± 1.122; *p* = 0.00), while the WOMAC total score, FJS, KSS, VAS and SF-12 were similar in both groups after surgery (Table 4 Figure 1).

### 3.4. Complications

After at least five years of follow-up, there were no differences in terms of the absolute number of complications. In the CAS group, there were two cases of aseptic loosening of the tibia that required knee replacement and a subtrochanteric fracture after falling, which required endomedullary osteosynthesis of the femur. In addition, one case of wound skin necrosis healed without further surgery. In the cTKA group, one case of tibial aseptic loosening and one case of septic loosening of both the femoral and tibial components required knee replacement. In addition, there was an intraoperative fracture of the external femoral condyle, which did not avoid knee implantation.

### 3.5. Outliers

Postoperatively, regarding outliers, as defined by a coronal alignment of >±3°, there were 15 cases in the CAS group and 34 in the cTKA group. There were no significant preoperative differences between the groups. On the postoperative assessment, there were differences in the SF12-MCS (52.65 ± 5.31 vs. 51.82 ± 2.86; *p* = 0.002), WOMAC (16.96 ± 9.73 vs. 18.11 ± 3.59; *p* = 0.021) and KSS (83.73 ± 8.67 vs. 84 ± 3.78; *p* = 0.031).

## 4. Discussion

In this independent retrospective study, we evaluated the clinical outcomes and radiological results of 119 patients who underwent TKA, either cTKA (59 patients) or CAS (60 patients), at our institution between 2015 and 2017 with a minimum follow-up of 5 years. Defendants of CAS suggest better alignment results (18); therefore, they infer better long-term functional outcomes and survivorship. However, recent long-term studies have reported slight or no differences between these two methods [11,18].

Panjwani et al. [19] conducted a meta-analysis and concluded that the CAS group had better outcomes in terms of KSS and WOMAC scores. In contrast, several authors stated that no relevant differences were found between the groups. Kim et al. found no difference between groups [20] for the WOMAC scale and other functional scores and activities, whereas Farhan-Alanie et al. [21] found no difference in the KSS and Oxford knee scores. At the same time, Ollivier et al. [10] found no difference in the SF-12, FJS, KSS or knee injury and osteoarthritis outcome score (KOOS). This is consistent with the findings of our study, since all WOMAC, KSS, FJS, SF-12 and FJS scales were comparable between the groups at 4 years follow-up. Interestingly, when the sub-items of the WOMAC were evaluated, there was less pain punctuation in the CAS group, which was a finding that did not coincide with the VAS results or the PCS item of the SF-12, which might represent WOMAC scale issues with this item [22].

Outliers, defined as those with a coronal angle > 0 ± 3°, demonstrated better behavior on the KSS and WOMAC scales in the CAS group and a lower prevalence than in the cTKA group. Improvements in implant positioning using the navigation technique have been widely discussed [23], and it seems reasonable to acknowledge that link; because CAS allows intraoperative implant positioning assessment, it may prevent mistakes during implantation and therefore coronal malalignment alteration. In our population, there was no difference in the coronal malalignment, but there were better functional results for CAS patients, which might be due to its improvement in sagittal, rotation and slope alignment not measured in this study [24].

Radiographically, there were no differences in the alignment or posterior condylar offset. While there was a slight increase in preoperative coronal deformity, there were no differences in preoperative KL or postoperative alignment. Furthermore, discrepancies in the PCO ratio had no impact on range of motion or knee function. Recent publications have questioned the measurement of both PCO and PCOr and have recommended MRI as a more valid and reproducible technique [25,26].

The increase in the incidence and size of notching in the CAS group was statistically significant. Lee et al. [13] found more prevalence of notching when TKA is performed by CAS, which could result in an increase in periprosthetic fractures. This might be for two main reasons [27]. First, the CAS system uses the whole mechanical axis of the femur while cTKAs only use the distal portion of the anatomical axis by placing the intramedullary rod. This mechanical axis is designed from the center of the femoral head to a certain point in the trochlear groove. The way this is placed may alter the anterior reference and thus notching prevalence. Second, the CAS system allows for size adjustment and the rotation of femoral components, which frequently leads to a controlled increase in femoral external rotation and increases the risk of notching. On the sagittal plane, the femur is a bowed bone, and an increase in bowing may alter the femur position and increase occurrence of notching [28]. Other authors have stated the need for a TKA system that offers multiple femoral component sizes to improve fitting and thus reduce the prevalence of notching [29]. Fortunately, there was no periprosthetic fracture during follow-up, which could be attributed to notching.

Gøthesen et al. [30] highlighted an increased risk of revision in the CAS group after 2 years follow-up, but Patrick et al. [31] recently published a survival rate of 92.9% for CAS and 95.6% for cTKA at 17 years follow-up, which was similar to data previously published by Kim et al. [32] at 12.3 years of follow-up. In the present study, there were comparable rates of aseptic loosening for both the tibial and femoral components with no difference in septic loosening after 5 years. While it cannot be ignored that a longer follow-up period reveals further differences, it seems reasonable to consider that similar results to those cited will be found.

Longer surgical times have been reported for CAS [32]. In our series, a mean increase of 12 min was observed, which might be greater than in other series [33]. It is important to note that we calculated the time from incision to wound closure; therefore, the time for CA setup, tests, and final checkups for every patient was included. Even with this in mind, surgeons are continuously improving and, with less of a learning curve, surgical time may be reduced.

The present study has some limitations. First, it was a retrospective study, and all preoperative data were collected from clinical records.

Although the population size in our study may seem relatively low, it has sufficient statistical power to compare both groups as a representation of larger populations. Early follow-up may have provided interesting data on clinical progression and patient satisfaction. However, relevant results for TKA appeared in the long run; as a result, a 5-year follow up seemed important enough to evaluate differences between techniques.

In addition, patients were evaluated using plain radiographs, while some authors recommend postoperative CT scans [34] to correctly evaluate coronal and sagittal alignment.

## 5. Conclusions

In summary, according to our results and those previously published, we found no benefit of CAS over cTKA in terms of survival, complications or pain with a mean 5-year follow up. Only a slight benefit was found for ROM but with no impact on functional scales. By contrast, an increase in the incidence of notching was observed in the CAS group, which may lead to future complications. Nevertheless, its use may aid clinical results in outliers and prevent them from occurring. Thus, it may be considered as a helpful technology that aids to control cutting steps and implant positioning in a way that is certainly not achievable by manual technique. Therefore, although two approaches are comparable, CAS might be considered as a more consistent technique that is less prone to failures.

## Figures and Tables

**Figure 1 jpm-13-01365-f001:**
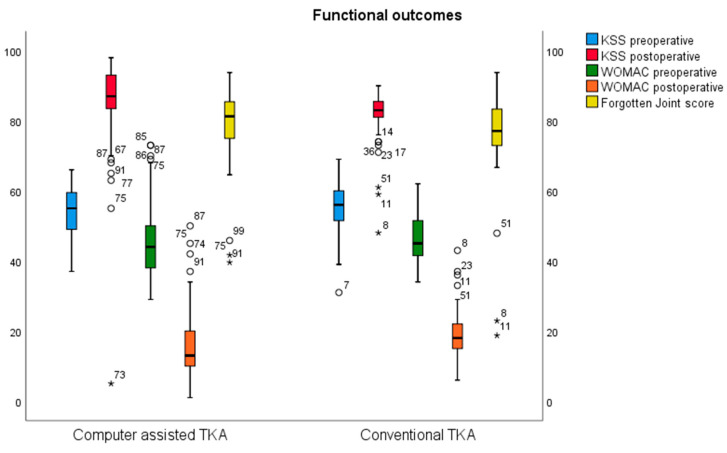
Functional outcomes. * Represents deviated values.

**Table 1 jpm-13-01365-t001:** Demographic data.

	Complete Sample N = 11	Computer-Assisted Group N = 60	Conventional TKA Group N = 59	t-Student *p*-Value
Weight (kg)	76.59 ± 9.00	76.72 ± 9.77	76.59 ± 9.00	*p* = 0.88 n.s
Height (cm)	164.50 ± 7.86	163.72 ± 6.63	164.50 ± 7.85	*p* = 0.07 n.s
IMC (kg/m)	28.37 ± 3.40	28.65 ± 3.59	28.36 ± 3.40	*p* = 0.35 n.s

**Table 2 jpm-13-01365-t002:** Radiographic results.

Variable	Sample N = 119	Computer-Assisted Group N = 60	Conventional TKA Group N = 59	t-Student *p*-Value
Hip–Knee–Ankle angle				
Preoperative	6.91 ± 6.34	7.37 ± 6.06	6.45 ± 6.61	*p* = 0.035 s.
Postoperative	2.27 ± 2.28	2.24 ± 2.32	2.29 ± 2.23	*p* = 0.87 n.s
Posterior condylar offset (PCO)				
Preoperative	0.42 ± 0.05	0.41 ± 0.05	0.43 ± 0.05	*p* = 0.83 n.s
Postoperative	28.08 ± 4.31	28.42 ± 5.28	27.73 ± 3.04	*p* = 0.39 n.s
Posterior condylar offset ratio (PCOr)				
Preoperative	0.45 ± 0.05	0.43 ± 0.59	0.46 ± 0.04	*p* = 0.12 n.s
Postoperative	0.45 ± 0.54	0.43 ± 0.06	0.43 ± 0.04	*p* = 0.00 s.
Notching				
Frequency	26.70%	40%	13.60%	*p* = 0.01 s.
Value (mm)	1.53 ± 1.92	1.24 ± 1.76	0.50 ± 1.42	*p* = 0.03 s.

**Table 3 jpm-13-01365-t003:** Clinical outcomes.

Variable	Sample N = 119	Computer-Assisted Group N = 60	Conventional TKA Group N = 59	t-Student *p*-Value
Flexion				
Preoperative	109.87 ± 11.48	113.08 ± 9.62	106.61 ± 12.37	*p* = 0.01 s.
Postoperative	115.42 ± 13.35	118.17 ± 14.08	112.63 ± 11.65	*p* = 0.02 s.
Extension deficit				
Preoperative	−1.64 ± 3.57	−0.5 ± 1.99	−2.8 ± 4.33	*p* = 0.09 n.s
Postoperative	0.31 ± 1.36	0 ± 0	−0.63 ± 1.88	*p* = 0.01 s.
Length of stay (Days)	5.97 ± 1.94	5.92 ± 1.83	6.02 ± 2.05	*p* = 0.78 n.s
Surgery duration (minutes)	101.50 ± 15.88	107.02 ± 15.22	95.32 ± 13.87	*p* = 0.00 s.
Hemoglobin				
Preoperative	14.29 ± 1.46	14.36 ± 1.40	14.21 ± 1.52	*p* = 0.01 s.
Postoperative	11.55 ± 1.48	11.74 ± 1.58	11.356 ± 1.363	*p* = 0.16 n.s
Hematocrit				
Preoperative	42.72 ± 4.08	42.83 ± 3.82	42.61 ± 4.36	*p* = 0.02 s.
Postoperative	34.22 ± 4.23	34.65 ± 4.33	33.78 ± 4.11	*p* = 0.26 n.s

**Table 4 jpm-13-01365-t004:** Functional outcomes.

Variable	Sample N = 119	Computer-Assisted Group N = 60	Conventional TKA Group N = 59	t-Student *p*-Value
KSS				
Preoperative	54.87 ± 7.01	54.43 ± 6.79	55.31 ± 7.25	*p* = 0.48 n.s
Postoperative	83.18 ± 11.18	84.67 ± 13.93	81.66 ± 7.24	*p* = 0.14 n.s
SF-12/PCS				
Preoperative	39.08 ± 4.76	38.53 ± 5.79	39.63 ± 3.38	*p* = 0.75 n.s
Postoperative	47.66 ± 6.59	47.15 ± 7.49	48.18 ± 5.56	*p* = 0.40 n.s
SF-12/MCS				
Preoperative	47.39 ± 5.82	48.16 ± 6.59	46.60 ± 4.87	*p* = 0.07 n.s
Postoperative	51.19 ± 4.97	51.85 ± 5.85	50.53 ± 3.81	*p* = 0.15 n.s
WOMAC scale				
Preoperative total	45.89 ± 8.99	45.35 ± 10.76	46.44 ± 6.79	*p* = 0.51 n.s
Postoperative total	17.68 ± 8.76	16.33 ± 10.14	19.05 ± 6.89	*p* = 0.09 n.s
Pain preoperative	12.12 ±4.43	12.75 ±5.69	11.47 ± 2.48	*p* = 0.49 n.s
Pain postoperative	3.27 ± 2.75	2.77 ±4.37	3.78 ± 2.75	*p* = 0.01 s.
Stiffness preoperative	4.39 ±1.65	4.00 ± 1.95	4.80 ± 1.16	*p* = 0.02 s
Stiffness postoperative	1.59 ± 1.34	1.17 ± 1.42	2.02 ± 1.12	*p* = 0.00 s.
Functional preoperative	29.38 ± 7.49	28.60 ± 8.91	30.17 ± 5.68	*p* = 0.47 n.s
Functional postoperative	12.82 ± 5.98	12.4 ± 6.95	13.25 ± 4.83	*p* = 0.44 n.s
Forgotten Joint score (FJS)	77.56 ± 11.89	78.96 ± 10.68	76.13 ± 12.94	*p* = 0.42 n.s
VAS score				
Preoperative	7.68 ± 1.19	8.08 ± 0.98	7.27 ± 1.26	*p* = 0.00 s.
Postoperative	2.12 ± 1.83	2.08 ± 1.93	2.15 ± 1.75	*p* = 0.84 n.s

## Data Availability

Data will be provided by the corresponding author upon request.

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
