# Peer review of "Clinical and Radiological Outcomes of Computer-Assisted versus Conventional Total Knee Arthroplasty at 5-Year Follow-Up: Is There Any Benefit?"

_jpm, 2023, doi:10.3390/jpm13091365_

Round 1

Reviewer 1 Report

The authors conduct an important analysis comparing outcomes from computer assisted total knee arthroplasty and conventional manual approach.

The results show that, in terms of outcomes, the two approaches are comparable. It is also found that the computer-based technique requires significant more time, but results to be more consistent with the results.

The study is highly relevant for the healthcare community addressing the emerging use of robots for orthopaedic surgery.

The work is well organized and clearly written.

I have only minor suggestions:

1. report the decimal numbers in the table with '.' rather than ','

2. change unpaired-t test with 2-sample t-test

3. I would suggest you emphasize more the fact that, despite the fact that the outcomes of the two approaches are comparable, computer assisted surgery is more consistent, providing a lower number of failures. I believe that, in general, this is the most important reason why computer assisted surgery would be recommended

Author Response

  1. Report the decimal numbers in the table with '.' rather than ',':

As you may see on the manuscript, I’ve already change all decimal numbers in the table.

  1. Change unpaired-t test with 2-sample t-test

Already assessed in the main text (Line 98)

  1. I would suggest you emphasize more the fact that, despite the fact that the outcomes of the two approaches are comparable, computer assisted surgery is more consistent, providing a lower number of failures. I believe that, in general, this is the most important reason why computer assisted surgery would be recommended

I add some words to conclusions based on your suggestions (Lines 236-239): “Thus, it may be considered as a helpful technology that aids to control cutting steps and implant positioning in a way that is certainly not achievable by manual technique. Therefore, although two approaches are comparable, CAS might be considered as a more consistent technique, less prone to failures.”

Reviewer 2 Report

I would like to thank the authors for the opportunity to review the manuscript entitled 'Clinical and radiological outcomes of computer-assisted versus conventional total knee arthroplasty at 5-year follow-up.' Is there any benefit?’ submitted for publication in JPM, MDPI. The Introduction provided a satisfactory background for the readers. The structure of the submitted article follows the standard IMRaD format (Introduction, Methods, Results, and Discussion). The manuscript includes all results relevant to the hypothesis. The general assessment is that this study can be recommended for publication after minor editing and addressing all thecomments presented below.

Specific Comments:

Comment 1: What was the design of the posterior stabilised implant used? Was it the same in all patients? 54-56

Comment 2: Write which computer-assisted surgery (CAS) system you use. Line 57

Comment 3: Provide citations for the Spanish version of the patient-reported outcome measures (PROM) you used. Lines 86-91

Comment 4: Which non-parametric test did you use to compare continuous variables without normal distribution? Lines 96-101

Comment 5: Is subtrochanteric fracture a complication of TKA? Lines 145-146 

Author Response

  1. Comment 1: What was the design of the posterior stabilised implant used? Was it the same in all patients? 54-56

For conventional group, we use Stryker-Triathlon® posterostabilised cemented implant while for CAS group, the chosen implant was Braun-Aesculap VEGA posterostabilised cemented implant” (Lines 56-58).

Both are cemented posterostabylised implants and had proven their results in previous literature. Both offer up to 8 sizes, to improve fitting and there’s currently no major differences in results available in the literature. We believe design impact on our sample outcome measurement is not determining.

  1. Comment 2: Write which computer-assisted surgery (CAS) system you use. Line 57

The CAS system was “Orthopilot navigator system TKA 5.1 version, Braun-Aesculap, Tuttlingen, Germany” (Lines 57-58). It has been modified in the manuscript.

  1. Comment 3: Provide citations for the Spanish version of the patient-reported outcome measures (PROM) you used. Lines 86-91

I’ve added citations to all PROMS (Line 89-93). Regarding FJS-12, there’s no specific literature in Spanish but is has been available and linguistically validated to Spanish since the very beginning.

  1. Comment 4: Which non-parametric test did you use to compare continuous variables without normal distribution? Lines 96-101

Further explained and corrected on lines 98-101. “Postoperative results were compared using an two-sample student’s t-test, with the assumption of homogeneity of variance used as appropriate, for quantitative and normal distributed variables; and Mann-Wthney U-test and Chi-square test for no parametric variables”

  1. Comment 5: Is subtrochanteric fracture a complication of TKA? Lines 145-146 

Subtrochanteric fracture is not a specific complication of TKA. Nevertheless, it affected this particular patient's PROM and also affected final HKA measurements.
